# Wrist Proprioception in Adults with and without Subacute Stroke

**DOI:** 10.3390/brainsci13010031

**Published:** 2022-12-23

**Authors:** Brittany M. Young, Rishika Yadav, Shivam Rana, Won-Seok Kim, Camellia Liu, Rajan Batth, Shivani Sakthi, Eden Farahmand, Simon Han, Darshan Patel, Jason Luo, Christina Ramsey, Marc Feldman, Isabel Cardoso-Ferreira, Christina Holl, Tiffany Nguyen, Lorie Brinkman, Michael Su, Tracy Y. Chang, Steven C. Cramer

**Affiliations:** 1Department of Neurology, University of California, 710 Westwood Plaza, Los Angeles, CA 90095, USA; 2California Rehabilitation Institute, 2070 Century Park East Rm 117, Los Angeles, CA 90067, USA; 3Department of Rehabilitation Medicine, Seoul National University College of Medicine, Seoul National University Bundang Hospital, 82 Gumi-ro 173 beon-gil, Bundang-gu, Seongnam-si 13620, Gyeonggi-do, Republic of Korea

**Keywords:** proprioception, stroke, rehabilitation, neurorehabilitation, sensory, recovery

## Abstract

Proprioception is critical to motor control and functional status but has received limited study early after stroke. Patients admitted to an inpatient rehabilitation facility for stroke (n = 18, mean(±SD) 12.5 ± 6.6 days from stroke) and older healthy controls (n = 19) completed the Wrist Position Sense Test (WPST), a validated, quantitative measure of wrist proprioception, as well as motor and cognitive testing. Patients were serially tested when available (n = 12, mean 11 days between assessments). In controls, mean(±SD) WPST error was 9.7 ± 3.5° in the dominant wrist and 8.8 ± 3.8° in the nondominant wrist (*p* = 0.31). In patients with stroke, WPST error was 18.6 ± 9° in the more-affected wrist, with abnormal values present in 88.2%; and 11.5 ± 5.6° in the less-affected wrist, with abnormal values present in 72.2%. Error in the more-affected wrist was higher than in the less-affected wrist (*p* = 0.003) or in the dominant (*p* = 0.001) and nondominant (*p* < 0.001) wrist of controls. Age and BBT performance correlated with dominant hand WPST error in controls. WPST error in either wrist after stroke was not related to age, BBT, MoCA, or Fugl-Meyer scores. WPST error did not significantly change in retested patients. Wrist proprioception deficits are common, bilateral, and persistent in subacute stroke and not explained by cognitive or motor deficits.

## 1. Introduction

Stroke remains a leading cause of disability and neurologic deficit worldwide [1,2]. Motor deficits are a major contributor to this burden of disability and have been substantially studied. Somatosensory deficits have received less study, but are also common: as many as 85% of patients with acute unilateral stroke report somatosensory deficits [3], and approximately 40% of patients with stroke report a sensory deficit that affects the upper extremity [4]. These common deficits in somatosensory function after stroke have additional functional implications, having been associated with reduced motor recovery [5,6,7,8] and poorer quality of life [9] following stroke. Therefore, evaluating somatosensory function and its subdomains remains an important goal, as proper identification of deficits can guide treatment and may provide valuable prognostic information [10].

The recovery of somatosensory deficits in the subdomain of proprioception is of particular interest in the setting of post-stroke recovery. Deficits in proprioception are found in the majority of subjects with stroke [11,12,13], can be present in both limbs after a unilateral infarct [5,12], and are associated with poorer outcomes [10,11,12] including reduced motor recovery [5,6,7] and poorer quality of life [9]. Motor recovery after stroke is thought to occur through a combination of spontaneous recovery and motor learning [14]. Sensory prediction error, which is informed by proprioceptive feedback, is a critical component of implicit motor learning [15]. Training-mediated recovery after stroke relies on motor learning and appears to be greatest in the first six months of stroke recovery [14,16]. There is an established association between impairments in proprioception and reduced motor recovery [5,6,7,17], which are likely due, at least in part, to limitations in motor learning which then limit the extent of motor recovery achieved by patients after stroke. Proprioceptive function in chronic stroke patients can also be predictive of subsequent treatment-induced motor gains [6,18]. Detailed quantification of proprioceptive deficits has been examined in chronic stroke [5,6,18]; however, precise evaluation of proprioceptive function has been less well studied in the early subacute period during the initial weeks following stroke onset.

The study of proprioception after stroke has historically been limited by methods of measurement that are either non-specific (e.g., scoring proprioception makes concomitant motor or tactile sensory demands, or may fail to differentiate the limb or laterality of any proprioceptive deficits that are identified), non-quantitative, or too technically demanding to be readily clinically implemented [19]. Some examples of validated assessments that have been used to assess proprioception in earlier literature include the Revised Nottingham Sensory Assessment [20], which requires sufficient motor control in the contralateral limb for the subject to actively mirror the passively manipulated position of the tested limb, and the Rivermead Assessment of Somatosensory Performance [21], which assesses proprioception by asking the subject to provide a yes/no responses to whether they feel a joint being passively moved and an up/down response regarding the direction of perceived movement. Another approach to evaluating proprioception is robotic position-matching [6,8], where subjects are asked to indicate when a specific position is reached during the course of a continuous passive movement. Other strategies have included the Thumb Localizing Test [22], in which the examiner passively positions the subject’s thumb in space with the subject asked to use their contralateral hand to grasp the passively positioned thumb, and manual testing, such as with the Sensory Subscale of the Fugl-Meyer Assessment [23], where a clinician moves a joint to a passive position and asks the subject to identify the direction of movement (e.g., up/down) or if the position matches one of a small number of pre-defined positions [24]. All proprioceptive assessments described here are performed with the subject unable to view the limb being tested.

Key longitudinal studies of somatosensory function after stroke have documented the greatest improvement in deficits during the first few months of recovery [12,25], though these studies used measures that were neither precisely quantitative nor specific to the proprioceptive domain. In contrast, the Wrist Position Sense Test (WPST), developed by Carey and colleagues [24,26], allows for the quantitative assessment of wrist proprioception in a clinical setting. This instrument has been validated in patients recovering from stroke as well as in healthy adults [24,26].

The current study aimed to characterize wrist proprioceptive deficits using the WPST in patients in the early subacute period post-stroke who were admitted to a U.S. inpatient rehabilitation facility, comparing results with healthy aging controls. To better interpret WPST findings, results were further examined for relationships with measures of demographics, upper extremity function, and cognition.

## 2. Materials and Methods

### 2.1. Subject Recruitment

Patients with stroke admitted to California Rehabilitation Institute, an inpatient rehabilitation facility (IRF), were recruited, as were aging healthy control subjects. Full eligibility criteria appear in Table 1. This study was approved by the local Institutional Review Board (University of California Los Angeles Medical IRB 3 Protocol #21-000936), and informed consent was obtained by a licensed physician from all subjects involved in the study. Subjects were required to have sufficient cognitive functioning to provide their own inform consent (i.e., no surrogate consent was used). All subjects were enrolled between January 2022 and August 2022. A sample size calculation was performed based on prior normative work by Carey and colleagues [26], using an estimated difference in WPST error between healthy controls and subjects with stroke of 8.4° with an estimated pooled standard deviation of 8°. This provided a minimum sample size estimate of 15 individuals in each group in order to detect a difference in WPST performance between the more-affected wrists of individuals recovering from stroke and that of healthy controls with 80% power at a confidence level of 95%.

### 2.2. Subject Assessments and Assessment Schedule

All subjects completed an initial set of assessments, referred to here as Visit 1. During Visit 1, all subjects were tested with the WPST according to standard administration practice as previously described [24,26]. Briefly, the subject was seated in a chair or (for subjects recovering from stroke) in a wheelchair selected by their IRF clinical care team that matched the body size and support needs of the patient. One forearm was placed into a tabletop cradle, ulna-side down. The cradle was hinged below the wrist joint, allowing the examiner sitting at the opposite end of the table to move the subject’s hand and thereby produce wrist flexion and extension in the plane perpendicular to the force of gravity. The position of the wrist was tracked along a large protractor fitted around the hinged cradle (the “lower protractor”). The lower protractor and the position of the subject’s wrist were visible to the examiner only. An opaque board displaying an analogous set of protractor positions (the “upper protractor”) and a moveable pointer were fitted above the subject’s wrist, obscuring the subject’s view of their distal forearm to their fingertips. A set of photographs demonstrating the positioning of the subject, examiner, and WPST apparatus is provided in Figure 1.

Figure 1 Wrist Position Sense Test Apparatus In Use.

The subject’s wrist was then passively moved from a neutral position to a second position ranging from 65 degrees of flexion to 125 degrees of extension. Subjects were instructed to use the non-tested upper extremity to move the pointer on the upper protractor to a point between 0–180 (with 90 representing the starting (neutral) position, i.e., with no flexion or extension of the wrist) to indicate the angle corresponding to where they felt their tested wrist was now positioned. Subjects who were unable to use the non-tested upper extremity to move the pointer in this way (e.g., subjects with significant motor deficits in the non-tested upper extremity) were asked to state aloud the numbered position on the upper protractor corresponding to where they felt the tested wrist was positioned. After successful completion of an unscored test trial, each wrist was passively moved through a series of 20 predetermined test positions using the WPST apparatus, with the subject asked to indicate their wrist position as outlined above for each test position. Each trial tested a single test position from this series. For each trial, the metric extracted from WPST testing is the number of degrees of error between the position to which the examiner actually moved the wrist and the position that the subject indicated using the upper protractor. This was recorded for each of the 20 test trials, separately for each wrist. The average amount of error across the 20 trials was then calculated for each wrist, with this average error value used as the WPST average error metric previously described by Carey and colleagues [24,26].

Subjects in the Control Group underwent WPST testing in their dominant wrist first, while subjects in the Stroke group had their less-affected wrist tested first; when no paresis was present, subjects in the Stroke group had the non-dominant wrist tested first. When feasible, a second set of assessments (Visit 2) was obtained from patients with stroke prior to discharge.

Additional data were collected at study visits. After recording basic demographics, at Visit 1, subjects were evaluated with the Box and Blocks Test (BBT) [27], this study’s primary measure of motor function. The Box and Blocks test was chosen as the primary measure of upper extremity motor function because it is in the activities limitation domain within the World Health Organization International Classification of Function [28], has been validated both in healthy aging subjects [27] and in patients recovering from stroke [29], is easily administered at bedside in a clinical setting, engages the sensorimotor system in a complex repetitive task, may present fewer floor effects among impaired individuals than assessments requiring a higher level of fine motor dexterity such as the Nine Hole Peg Test (NHPT), and in healthy controls does not have a maximum score that defines normal and so has reduced ceiling effect. As part of a battery of secondary motor and cognitive assessments, subjects were also evaluated using the NHPT [30,31], the Trail Making Test A [32,33], and the Montreal Cognitive Assessment (MoCA) [34,35] at Visit 1. Subjects in the Stroke Group were additionally evaluated at Visit 1 using the Fugl-Meyer Upper Extremity (FM-UE) Motor Assessment [36] and the National Institutes of Health Stroke Scale [37]. Assessments completed at Visit 2 included the FM-UE, WPST, BBT, and NHPT.

### 2.3. Statistical Analysis

Continuous and ordinal variables were tested for normality using Shapiro–Wilkes testing. Normally distributed variables were analyzed using parametric approaches, while analyses using data that were not normally distributed and could not be transformed used non-parametric approaches. Demographic variables and assessment scores at the group level were compared between Stroke and Control Groups using Student’s *t*-test for normally distributed continuous and ordinal variables, using Wilcoxon rank sum test for continuous and ordinal variables not normally distributed, and using Fisher’s exact test for categorical variables. Within-subject assessment scores were also compared within-visit (e.g., dominant vs. nondominant hand) and across visit (i.e., Visit 1 vs. Visit 2) using the paired versions of Student’s *t*-test and Wilcoxon rank sum test for normally and non-normally distributed data, respectively.

To determine whether the 20 positions used in WPST testing vary in a systematic way or not during the progression from the beginning to the end of the preset sequence of test positions, a Spearman rank correlation analysis was used to assess for a potential relationship between the degree of deviation that the subject is moved from the neutral position (i.e., deviation from the position designated as “90°” in the WPST instrument) and the trial number (from 1 to 20) associated with each test position.

To assess for potential effects of position and trial number on the amount of error present in subjects’ responses to individual WPST trials, linear mixed effect modeling was used to evaluate the effect of trial number and degree of deviation from the neutral position on the amount of error observed at the level of individual trials. In these models, amount of error was the dependent variable, position deviation from neutral and trial number were each fixed factors in their respective models, and subject was specified as a random factor in each model.

Overall performance on the WPST (i.e., mean error, in degrees) at Visit 1 was evaluated in relation to age and scores on each Visit 1 behavioral assessment, separately for each subject group. Pearson correlation analysis was used for normally distributed data, and Spearman correlation analysis was used for non-normally distributed data.

The primary demographic variable of interest was age, and the primary behavioral variable of interest was upper extremity function (i.e., scores on the Box and Blocks test). The threshold for statistical significance for each was set at *p* < 0.05. Remaining analyses were corrected for multiple comparisons using false discovery rate correction [38] to produce corrected *p*-values.

## 3. Results

### 3.1. Subject Demographics

Subject demographics appear in Table 2. There were no differences between the Stroke and Control groups with respect to age, gender, or (pre-stroke) hand dominance.

For those in the Stroke Group, Visit 1 occurred within 11 days of admission to the IRF (4.9 ± 2.7 days, mean ± SD) and within 30 days of stroke onset (12.5 ± 6.6 days, mean ± SD). Some individuals (n = 12) in the Stroke Group completed a second set of assessments (Visit 2) shortly before discharge from the IRF. For these subjects, the time between Visit 1 and Visit 2 averaged 10.6 ± 7.4 (range 6–29) days. For those in the Control Group, all Visit 1 testing was completed on the same day.

### 3.2. Assessment Performance

Assessment scores appear in Table 3 for the Control Group and Table 4 for the Stroke Group. During initial WPST administration, all subjects in the Control Group indicated a response consistent with wrist deviation in the same direction as the initial test position when presented with an unscored test trial for each wrist, supporting appropriate understanding of assessment instructions. Similarly, all subjects in the Stroke Group indicated a response consistent with wrist deviation in the same direction as the initial test position when presented with an unscored test trial for the first wrist tested (i.e., in the less-affected wrist). Three subjects in the Stroke Group indicated a position with deviation in the opposite direction from the initial test position specifically when the more-affected wrist was tested. However, these subjects had been able to demonstrate appropriate understanding of the task instructions during testing of the less-affected wrist and were also able to verbally confirm understanding of the task instructions when questioned further after the unscored test trial of the more-affected wrist.

Using the 95th percentile cutoff of at least 9.5° of average error as the threshold for identifying proprioceptive impairment [26], 7 of the 19 subjects in the control group had proprioceptive impairment in the dominant upper extremity, and 7 of the same 19 subjects had proprioceptive impairment in the nondominant upper extremity. Using the same criteria, 15 of 17 subjects in the Stroke Group had proprioceptive impairment in the more-affected upper extremity (one subject was unable to complete the WPST with the more-affected upper extremity due to shoulder subluxation precluding appropriate positioning), and 13 of the 18 subjects in the Stroke Group had proprioceptive impairment in the less-affected upper extremity. Proprioceptive impairment was more common in patients with stroke, as compared to healthy controls, in the more-affected (*p* = 0.002) and less-affected (*p* < 0.05) upper extremity (Fisher’s exact test).

### 3.3. Trial and Position Effects on WPST Testing

In the non-dominant wrist of Control Group subjects, later trials (higher trial number, *p* = 0.003) and trials nearer to end range of motion (test position has higher deviation from the neutral position, *p* = 0.018) were each independently associated with larger error. Similarly, higher trial number (*p* = 0.008) and higher deviation from the neutral position (*p* = 0.003) were also associated with increased error in the more-affected wrist of Stroke Group subjects. These effects were not found in the dominant wrist of Control Group or the less-affected wrist of Stroke Group subjects, nor in either wrist of Stroke Group subjects who completed Visit 2 testing. In terms of the structure of the WPST test, trial number (from 1 to 20) was not related to the degree of deviation from neutral position—late trials are not closer or further from end range of motion positions as compared to early trials.

### 3.4. Within-Subject Differences

There were no significant differences in performance between the dominant and non-dominant upper extremity in Control Group subjects for the WPST (*p* = 0.4, Figure 2), NHPT (*p* = 0.4), or BBT (*p* = 0.8). In contrast, at Visit 1 subjects in the Stroke Group demonstrated significantly worse performance in the more-affected upper extremity as compared to the less-affected upper extremity on the WPST (*p* = 0.019, Figure 2), NHPT (*p* = 0.019), and BBT (*p* = 0.034). At Visit 2, no such differences were present (*p* > 0.05 for each assessment). Subjects who completed testing at both Visit 1 and Visit 2 did not demonstrate any significant differences between the two visits in WPST, NHPT, BBT, or FM-UE scores (*p* > 0.1 for each assessment).

### 3.5. Group Comparisons

Subjects in the Stroke Group performed significantly more poorly than those in the Control Group in all behavioral assessments (*p* < 0.001). Figure 2 shows WPST error, which in the more-affected wrist of Stroke Group subjects was higher than in the dominant (*p* = 0.0012) or non-dominant (*p* = 0.0002) wrist of Control Group subjects, while WPST error in the less-affected wrist did not differ from either wrist of Control Group subjects (*p* > 0.05).

### 3.6. Demographic and Assessment Relationships with WPST Performance

WPST error was related to age and behavior in Control Group subjects but not Stroke Group subjects. In Control Group subjects, higher WPST error in the dominant wrist was associated with higher age (r = 0.54, *p* = 0.018, Figure 3A) and poorer BBT performance (r = −0.60, *p* = 0.006, Figure 3B); relationships with longer Trail Making Test A time (r = 0.55) and longer NHPT time (r = 0.48) were suggested but did not survive correction for multiple comparisons. No significant relationships were identified between any assessment and (a) WPST error in the nondominant wrist of the Control Group or (b) WPST error on either side among subjects in the Stroke Group.

Figure 3 Correlations Between Age and Dominant Upper Extremity Box and Blocks Scores with Wrist Position Sense Test Performance in the Dominant Wrist of Healthy Aging Individuals

## 4. Discussion

Although persistent sensory deficits are common among patients with a history of stroke [25] and may influence post-stroke motor recovery [6,7,18,19] as well as functional status [10], detailed characterization of proprioceptive function has been limited in this population. Proprioceptive function has also been shown to differ across the lifespan in healthy individuals [8]. The work described here investigated proprioceptive function assessed using the Wrist Position Sense Test [24,26] in individuals specifically in the early subacute stage of stroke recovery and in aging adults with no stroke history. These findings underscore the prevalence of proprioceptive deficits after stroke as well as the subclinical decline in proprioceptive function with age. This work also underscores the importance of assessing sensory deficits in patients after stroke, as these deficits may not be captured by assessments in other behavioral domains.

### 4.1. Deficits in Proprioception Were Common after Stroke

Among patients with early subacute stroke, nearly all (15 of 17, 88%) demonstrated proprioceptive deficits in their more-affected wrist, and the majority (13 of 18, 72%) also showed proprioceptive deficits in their less-affected wrist. WPST scores were stable, as they did not significantly change in the short interval from Visit 1 to Visit 2 among retested subjects, though no subject received any intervention that specifically targeted the proprioceptive or somatosensory systems between these two visits. Prior longitudinal studies of proprioceptive function after stroke reported far lower prevalence of proprioceptive deficits (<30%), even in the acute to early sub-acute stages of stroke recovery [12,25]. However, these studies estimated proprioception using gross bedside scales (Nottingham Sensory Assessment and Rivermead Assessment of Somatosensory Performance, respectively) that are not specific to the upper extremity. In contrast, prior work using the WPST found that 60% of patients with recent stroke had proprioceptive deficits [24]. Other work using robotic technology for a precise, quantitative approach to assessing proprioception in the upper extremity found a similar prevalence of impaired proprioceptive function (67%) among patients in the chronic stage of stroke [5,18]. The findings presented here agree with the latter studies, reinforcing that proprioceptive deficits are highly prevalent after stroke and are best captured using a precise, quantitative testing instrument. The finding that proprioceptive deficits are common early after stroke may inform strategies for therapies targeting sensory or motor recovery.

### 4.2. Proprioception Deficits in Controls Increased with Age

Proprioceptive deficits were encountered in 7 out of 19 (37%) control subjects, for each wrist, a rate that was higher than expected given that the threshold defining these impairments was previously established at the 95th percentile of WPST performance from a sample of healthy subjects [26]. Higher age was also found to correlate significantly with greater WPST error in the current cohort of healthy subjects, which also contrasts with the prior WPST validation study [26], where age was not found to correlate with WPST error [26].

These differences may be due to differences across studies in the age range examined. The study that validated WPST performance [26] enrolled healthy subjects ranging from 21–79 years in age, with a mean age of 52 years. In contrast, age in the current cohort of healthy subjects ranged from 60–95 years, with a mean age of 74 years, as such subjects more closely approximate the average age of stroke onset. Similar to the current findings, Ingemanson and colleagues have previously documented an age-related decline in passive finger proprioceptive function in a cohort of healthy subjects ranging in age from 22–87 years old [8]. This age-related decline was specific to test conditions in which visual input was unavailable, as is true for the WPST. Taken together, these results suggest that age-related effects may be subtle (as estimated in this work to account for <0.2° of average error per year of additional age) but are more likely to be detected among elderly healthy subjects, i.e., those with age above the upper age limit from the prior WPST validation study [26].

### 4.3. Proprioceptive Error Increased over a Subject’s Course of Testing and as Positions Approached End Range of Motion

WPST raw error increased with higher trial number and with larger deviation from a neutral (neutral defined as no flexion or extension) position when testing the nondominant wrist in the Control Group and the more-affected wrist in the Stroke Group. These effects were small, estimated to range from 0.19–0.36° of additional error per additional trial and from 0.05–0.14 additional degrees of error per additional degree of deviation tested, respectively. It is unclear why these effects were only seen in one wrist among both groups, particularly given that those in the Control Group showed no significant difference in overall WPST performance between the dominant and nondominant wrists.

Noting that these effects were appreciated only in the second wrist to be tested within each study group, it is possible that test fatigue contributed to the emergence of these effects during the second half of WPST administration in both groups. Prior work has shown reproducible declines in cognitive task performance with prolonged repeated administration among young healthy individuals, though these effects appear to be attenuated in older healthy individuals [39]. Multiple studies have also shown reduced proprioceptive function and postural control following physical fatigue [40,41], even when this fatigue is generalized [42] or present specifically in muscles remote from those being evaluated for proprioceptive function [43]. In this context, subjects may have had reduced cognitive or attentional reserve when completing the latter half of WPST assessment, allowing for greater error with additional trials or when tested on positions further from neutral. As there was no inherent relationship between trial number and degree of deviation from neutral position, the effects of test position and trial number on WPST error most likely represent two independent effects.

### 4.4. Proprioceptive Deficits in the Dominant Wrist of Healthy Subjects Were Related to Motor Function

The sensory and motor systems function in a highly integrated manner to learn and produce skilled movements in healthy humans [44]. Proprioception is a key sensory domain of particular interest in relation to motor function [44,45] and motor learning [15], which rely in part on feedback provided by proprioceptive inputs during learned and novel motor behaviors. Increased noise in sensory feedback also slows motor adaptation in healthy older adults [46]. In this work, the Control Group showed a significant relationship between higher WPST scores in the dominant wrist with higher BBT scores using the dominant upper extremity. Although these subjects were not provided with an opportunity to practice or serially tested to assess formal motor learning, this relationship was somewhat expected given the importance of proprioception in performing everyday motor tasks requiring reaching, grasping, and manual manipulation of objects in space—all elements of the movements required to perform the BBT. Other work investigating proprioceptive and motor functions in the lower extremities has similarly demonstrated lower hip proprioception error to correlate with better dynamic balance in aging adults [47].

What is less clear is why this relationship was noted only in the dominant hand of Control Group subjects. Given that a history of tool use [48] and proprioceptive memory [45] are also thought to influence the use of proprioceptive information when executing motor tasks in healthy individuals, the relationship between WPST and BBT performance may result from baseline differences in learned movements and a lifetime of tool use experience specific to the dominant upper extremity. Alternately, it may simply be that proprioceptive function and motor performance are not as strongly related in the nondominant upper extremity.

### 4.5. Proprioceptive Deficits Are Not Explained by Cognitive or Motor Deficits after Stroke

Subjects with stroke performed less well on multiple behavioral assessments as compared to aging control subjects; these included the TMTA, NHPT, and BBT, in the more-affected upper extremity; the latter two assessments also showed impaired performance using the less-affected upper extremity. These findings are consistent with prior work that has demonstrated impairments in both the more-affected and less-affected upper extremities after stroke [24,49]. Although subjects with stroke performed more poorly than controls using the less-affected hand on motor assessments such as the NHPT and BBT, these motor deficits did not extend to proprioceptive dysfunction, as there were no significant differences between WPST performance in the less-affected wrist of subjects with stroke and WPST performance of either hand in the Control Group.

In contrast to the Control Group in this study and to prior work that has shown proprioceptive deficits to correlate with scores on multiple motor assessments [50] and functional [25] outcomes, individual differences in WPST performance after stroke were not accounted for by age, motor function, performance on cognitive assessments, or by global burden of neurologic deficits in this cohort. A notable difference in findings between this work and that described in prior literature is the current specific focus on individuals in the early subacute stage of stroke. Sensorimotor relationships might vary at different timepoints following stroke onset. The use of a precise, quantitative measure of proprioception (i.e., the WPST) that limits (or for hemiplegic patients, eliminates) the need for ipsilateral motor or contralateral sensorimotor function for proprioceptive task completion is another notable difference between the current findings and prior literature. When working with a population where motor deficits are frequently encountered, having the option to use verbal responses to indicate perceived position among individuals with limited to no motor function in the untested wrist is a key strength of the WPST. If proprioceptive function is evaluated using a paradigm that requires functionality in additional domains, such as the thumb localization test [50], it becomes difficult to interpret proprioceptive scores and their change over time. The lack of an association between proprioceptive performance and demographic/behavioral measures in the Stroke Group further emphasizes the importance of scoring assessments that specifically target sensory deficits after stroke, as these deficits cannot be readily predicted or derived from deficits measured in other domains during the subacute stage of stroke recovery.

### 4.6. Limitations

A number of limitations are present. Enrollees more often had deficits on the left side, possibly due to difficulties in recruiting patients with aphasia. Although the current sample size was sufficient to detect some effects, cohorts were modestly sized and so some analyses may have been underpowered.

## 5. Conclusions

Proprioceptive function in the dominant upper extremity of healthy subjects may decline sub-clinically with advancing age and may be predicted by motor performance in the same upper extremity. In contrast, deficits in proprioception are common among patients after stroke but are not readily predicted by deficits in motor or cognitive function. These post-stroke deficits affect the less-affected as well as the more-affected upper extremity. Further quantitative studies of the proprioceptive system, including recovery of proprioceptive deficits and the factors that influence their severity, will provide a valuable dimension to the understanding of post-stroke recovery.

## Figures and Tables

**Figure 1 brainsci-13-00031-f001:**
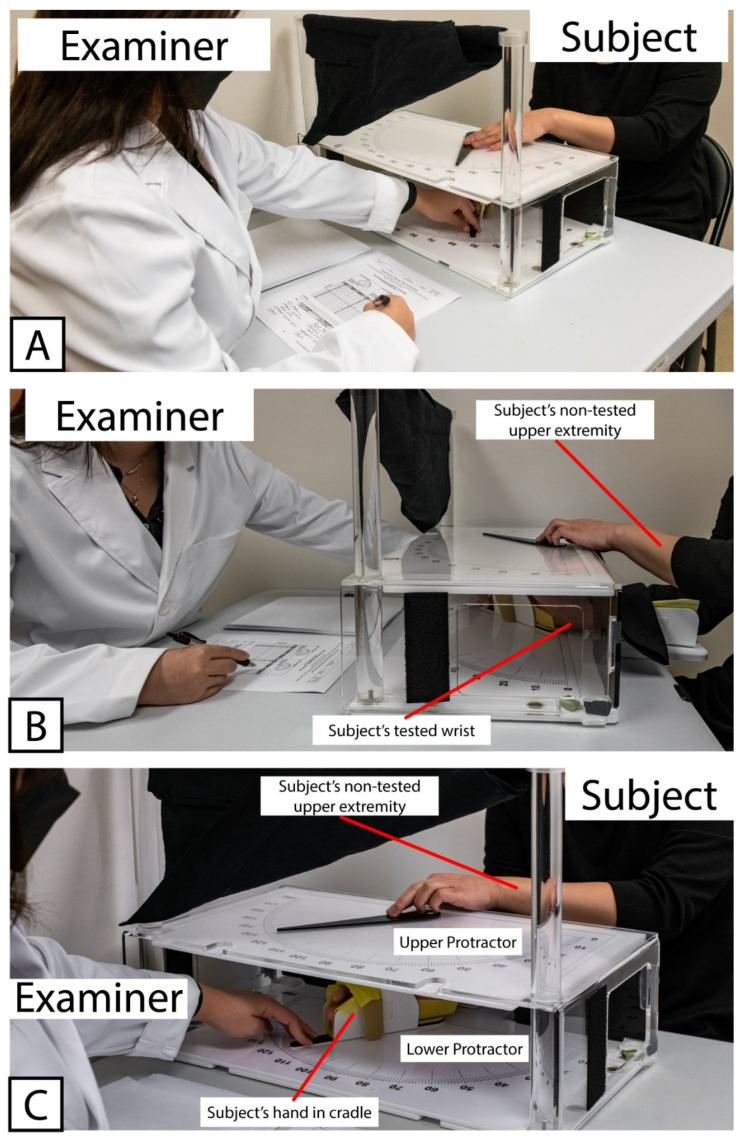
Example showing positioning of Examiner, Subject, and testing apparatus used for Wrist Position Sense Test administration. (**A**) Relative positioning of Examiner and Subject, with WPST Apparatus situated between them. (**B**) Positioning of Subject’s tested wrist within WPST apparatus, as well as non-tested upper extremity used to manipulate pointer on the top surface of WPST apparatus. (**C**) Relative placement of Upper and Lower Protractors. Note that the Upper Protractor obstructs the subject’s line of sight, preventing subject’s ability to view the tested wrist.

**Figure 2 brainsci-13-00031-f002:**
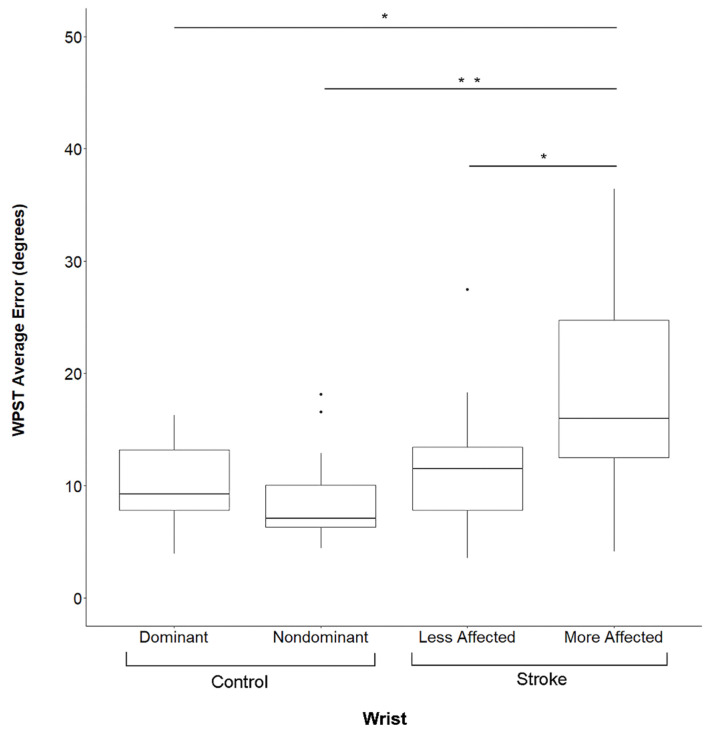
Wrist Position Sense Test Error By Subject Group And Side. Data are for Visit 1. WPST = Wrist Position Sense Test, * significant at *p* < 0.01 after correction for multiple comparisons, ** significant at *p* < 0.001 after correction for multiple comparisons, Dots represent outlier data points for respective box and whisker plots.

**Figure 3 brainsci-13-00031-f003:**
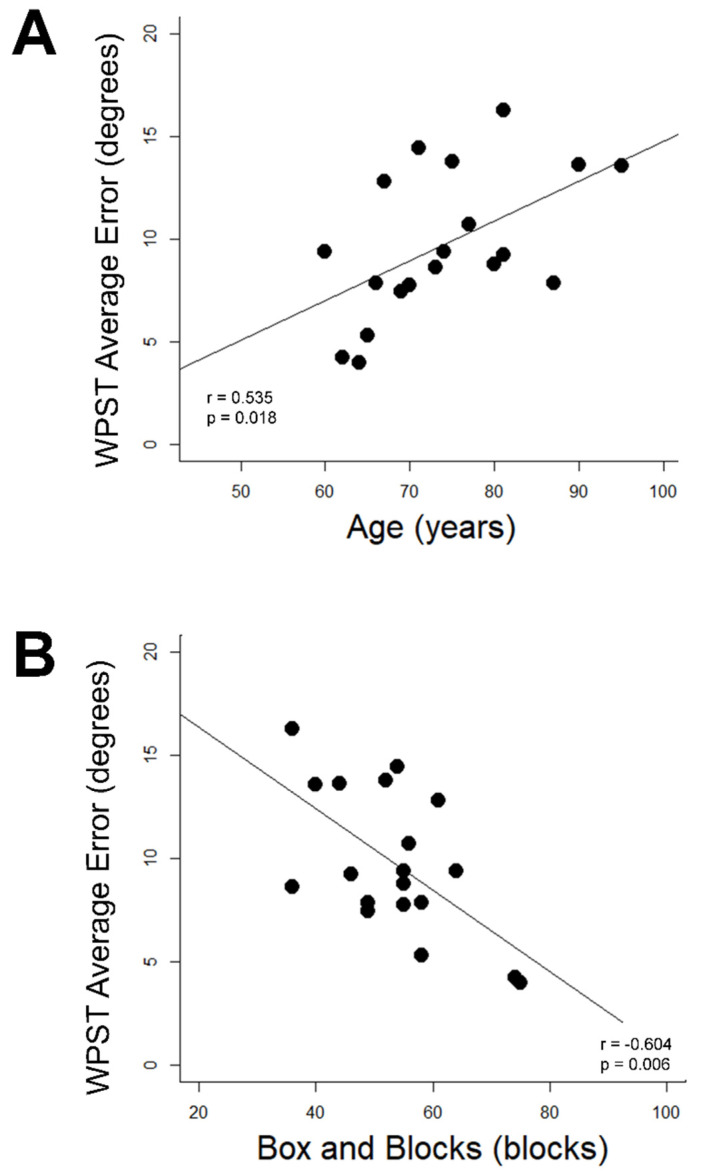
Correlates of dominant wrist WPST error in Control Group subjects. (**A**) WPST error correlates with higher age (**B**) WPST error correlates with Box and Blocks Test performance. WPST = Wrist Position Sense Test.

**Table 1 brainsci-13-00031-t001:** Study Eligibility Criteria.

Inclusion Criteria	Exclusion Criteria
Age ≥ 18 years old	A major, active musculoskeletal or peripheral nerve disease that significantly affects upper extremity function
2. Able to provide informed consent (i.e., no surrogate consent)	2. Deficits in cognition or communication that interfere with reasonable study participation
3. Admitted to California Rehabilitation Institute with a stroke with onset ≤ 30 days prior *	3. Lacking visual acuity, with or without corrective lenses, of 20/40 or better in at least one eye
	4. Non-English speaking, such that the subject does not speak sufficient English to comply with study procedures
	5. Expectation that the subject is unable or unwilling to perform study assessments
	6. Is on isolation precautions (contact, droplet, airborne, or modified for COVID-19)

* Applies only to patients with stroke.

**Table 2 brainsci-13-00031-t002:** Baseline Subject Demographics.

	Subjects with Stroke	Control Subjects	*p*-Value
**n**	18	19	
**Age (range), in years**	69.1 ± 16.3 (32–91)	74.1 ± 9.7 (60–95)	0.27
**Gender**	7 Female, 11 Male	14 Female, 5 Male	0.15
**Stroke-Affected Hand**	15 Left, 3 Right	Not Applicable	
**Time Post-Stroke (days between index stroke and Visit 1 testing)**	12.5 ± 6.6	Not Applicable	
**Dominant Hand**	14 Right, 1 Left, 2 Right-dominant ambidextrous	18 Right, 1 Left-dominant Ambidextrous	0.23

Mean ± SD. For subjects with stroke, dominant hand refers to pre-stroke.

**Table 3 brainsci-13-00031-t003:** Control Group Scores.

Assessment	Score
n	19
Montreal Cognitive Assessment	26.5 ± 2.0
Trail-Making Test A, time (sec)	28.7 ± 8.4
Wrist Position Sense Test Dominant upper extremity (degrees)	9.7 ± 3.5
Wrist Position Sense Test Non-Dominant upper extremity (degrees)	8.8 ± 3.8
Nine Hole Peg Test Time, Dominant Hand (sec)	22.7 ± 7.9
Nine Hole Peg Test Time, Non-Dominant Hand (sec)	22.9 ± 4.0
Box and Blocks Test Dominant Hand (# blocks in 60 sec)	53.5 ± 10.8
Box and Blocks Test Non-Dominant Hand (# blocks in 60 sec)	53.8 ± 11.0

Mean ± SD.

**Table 4 brainsci-13-00031-t004:** Stroke Group Scores.

Assessment	Visit 1 Score	Visit 2 Score
n	18	12
National Institutes of Health Stroke Scale	3 [2–7.5]	N/A
Montreal Cognitive Assessment	20.7 ± 5.1	N/A
Trail-Making Test A, time (s)	77.8 ± 35.4	N/A
Fugl-Meyer Upper Extremity Motor	44.9 ± 23.9	45.1 ± 23.4
Wrist Position Sense Test More-Affected Upper Extremity (degrees)	18.6 ± 9.0	17.1 ± 7.7
Wrist Position Sense Test Less-Affected Upper Extremity (degrees)	11.5 ± 5.6	15.7 ± 16.3
Nine Hole Peg Test Time, More-Affected Hand (s)	46.8 ± 12.6	46.7 ± 13.2
Nine Hole Peg Test Time, Less-Affected Hand (s)	35.1 ± 13.4	32.8 ± 12.1
Box and Blocks Test More-Affected Hand (# blocks in 60 s)	22.3 ± 16.5	21.2 ± 18.6
Box and Blocks Test Less-Affected Hand (# blocks in 60 s)	31.8 ± 10.9	34.6 ± 10.1

Mean ± SD or median [Inter-Quartile Range].

## Data Availability

The data presented in this study are available on request from the corresponding author. The data are not publicly available due to the decision to make available upon request.

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
