# Peer review of "Wrist Proprioception in Adults with and without Subacute Stroke"

_brainsci, 2022, doi:10.3390/brainsci13010031_

Round 1

Reviewer 1 Report

The subject of this article seems interesting to me, however there are some aspects that must be modified.

First of all, it is important to indicate the date of the study and provide the number of the ethics committee. Table 1 is not well inserted and makes it difficult to read, they should redo it for better reading.

It would be advisable to include a list of abbreviations to better follow the manuscript.

Authors should include a photograph of the position in which the intervention was performed to help the reader understand correctly and more visually how the procedure was performed.

How did the authors calculate the necessary sample in each group to carry out this study?

Table 3 is not correctly inserted, please correct it.

Reviewer 2 Report

This was a cross-sectional study that investigated proprioceptive impairment in stroke patients.  It is important to investigate the degree of proprioceptive impairment and related functions in stroke patients. However, the paper was unclear and many revisions were needed.

Abstract

Line 21 18.6 9° This is mistake.

Introduction

How does proprioceptive sensory impairment after stroke affect recovery of motor function? Please explain in more detail using previous research.

Why is it important to investigate proprioceptive sensory impairment in stroke patients?

And what tests are used to investigate proprioceptive sensory impairment in the chronic phase? Please describe more specifically.

Materials and Methods

2 of 15 lines 72

This is a cross-sectional study, but the sample size is small.

I assumed that this number is a case of high effect size in previous studies. How was the sample size calculated?

2 of 15 lines 80

In order to investigate proprioception, it is important that cognitive function be preserved.

However, it was unclear to what degree cognitive function was specifically excluded.

3 of 15 lines 82

It is difficult to read the methods. Please add diagrams or pictures to make it easier for the reader to understand.

6 of 15

TMT-A in the control group is longer than in stroke patients. This TMT-A value is attention deficit.

Why is attention deficit worse than in stroke patients?

Round 2

Reviewer 1 Report

Thank you.

Reviewer 2 Report

Thank you for the sufficient revision.